# Psychological Impact of Cancellation of Elective Surgeries for Ophthalmic Patients during COVID-19 Pandemic

**DOI:** 10.3390/ijerph192214852

**Published:** 2022-11-11

**Authors:** Stephanie K. Y. Chu, David T. C. To, Candice C. H. Liu, Tony Wong, Kenneth K. W. Li

**Affiliations:** 1Department of Ophthalmology, United Christian Hospital, Hong Kong; 2Department of Ophthalmology, Tseung Kwan O Hospital, Hong Kong; 3Department of Ophthalmology, School of Clinical Medicine, LKS Faculty of Medicine, The University of Hong Kong, Hong Kong; 4Department of Clinical Psychology, United Christian Hospital, Hong Kong

**Keywords:** COVID-19, cataract, psychosocial impact, cataract surgeries

## Abstract

The COVID-19 pandemic has disrupted routine hospital services globally. The cancellation of elective surgeries placed a psychological burden on patients. A questionnaire study was conducted to identify the psychological impact of canceling cataract operations on patients at Kowloon East Cataract Center, Tseung Kwan O Hospital, Hong Kong, from April to June 2020. In total, 99 participants aged 59 years old and above, who had their cataract surgeries postponed or as scheduled, were studied using the standardized patient health questionnaire (PHQ-9) and generalized anxiety disorder questionnaire (GAD-7). None of the patients who had their cataract surgeries rescheduled reached the cutoff score for major depression in PHQ-9, whereas, according to GAD-7, five patients had mild symptoms of anxiety, and one had severe symptoms. There was no significant psychosocial impact of the cancellation of cataract surgeries on patients.

## 1. Introduction

Cataract is one of the leading causes of visual impairment [1]. Visual impairment is reported to have a greater negative impact on the quality of life of the elderly than other age-related conditions, such as increased depression, social isolation, and increased mortality [2]. As a result of the emergence of coronavirus disease 2019 (COVID-19), there has been substantial disruption to hospital services [3,4]. Cataract surgeries were suspended and postponed as medical resources had to be reserved for urgent procedures amid the COVID-19 pandemic. For instance, there was a dramatic reduction in preventative and elective care [5]. Ophthalmological patients were also affected by an increase in waiting times, with limited access to cataract surgeries, especially in public-funded healthcare systems [6]. European countries recorded a 97% reduction in the number of cataract surgeries from March to April 2020, compared with the same period in 2019 [7]. Besides cataract surgeries, patients who suffered from other ophthalmological pathologies, such as glaucoma, experienced difficulties in acquiring medication, and reported the subjective worsening of their ocular condition [8]. As the pandemic continues to evolve, patients may display higher levels of depression and anxiety.

Although there are studies that consider psychological stress in frontline medical healthcare workers [9,10], the psychological burden on patients has not been studied. For instance, studies have shown that the postponement of cataract surgeries due to COVID-19 led to the progression of cataracts, showing the pandemic’s negative impact on patients’ management [11]. This shows that COVID-19 has caused significant physical effects on patients, yet the psychological effects are not explored. Depression affects around 5% of men and up to 10% of women in developed countries, whereas up to 33% of people might develop an anxiety disorder at some point in their life [12,13,14]. The increasing prevalence of these mood disorders is largely attributed to environmental factors, particularly stress [15]. The implementation of social distancing, contact tracing, and quarantine measures has been proven to influence both the psychological and physical health of the general population [16]. The cancellation of cataract surgeries and the progressive deterioration of visual function in patients has had a negative impact on their quality of life and psychological state. This study aims to identify the psychological impact of cancelling cataract operations on patients during the pandemic.

## 2. Materials and Methods

This is a prospective, observational cohort study. All consecutive cases that were scheduled to undergo cataract operations at Wu Ho Loo Ling Cataract Centre, Tseung Kwan O Hospital, Hospital Authority, Hong Kong SAR, China, from April to June 2020 were prospectively recruited for the study.

Patients were divided into two groups. The treatment group included patients with their surgeries postponed, while the patients in the control group underwent the surgeries as scheduled.

This research was approved by the Research Ethics Committee (Kowloon Central/Kowloon East) (Ref: KC/KE-20-0162/ER-2). The questionnaire study was clearly explained to all participants and informed consent was obtained prior to the study.

After obtaining informed consent, a survey comprising twenty-one questions was given to each patient. The survey included patients’ demographics, knowledge, and practices towards COVID-19. In order to avoid inflicting additional anxiety on patients before surgery, our trained staff conducted questionnaires for the composite measurement of depression and anxiety after the completion of surgeries. The patient health questionnaire (PHQ-9) and generalized anxiety disorder questionnaire (GAD-7) were used. The PHQ-9 total score for the nine items ranges from 0 to 27. Using the mental health professional (MHP) reinterview as the criterion standard, PHQ-9 score ≥10 was used as a cutoff for major depression [17]. For GAD-7, a score of 0–4 was considered to be minimal, 5–9 to be mild, 10–14 to be moderate, and 15–21 to be severe [18].

### Data Analysis

The inter-rater reliability of surveyors was used as there was more than one surveyor. The point prevalence was calculated. The prevalence was calculated by gender, age group, and preoperative visual acuity for subgroup analysis. Pearson Chi-square was used to test for statistical significance between each sub-group. The confidence interval for prevalence was calculated using R software version 3.6.3. To study the association between depression, anxiety status, and a set of possible explanatory variables, regression analysis and the Chi-square test were used. Odds ratios with confidence intervals were calculated in R software. A *p*-value less than 0.05 was considered to be statistically significant.

In total, 99 cases of 99 patients, with a mean age of 74.2 (range 55–89), were included in the study. There was no gender preponderance (56 versus 43). All patients were of Chinese ethnicity. Demographics of patients are summarized in Table 1.

## 3. Results

During the COVID-19 outbreak, a significant number of elective cataract surgeries were postponed due to service reductions. Among the 99 patients, 74 had their surgeries postponed (Group 1) while 25 did not (Group 2).

The PHQ-9 and GAD-9 results are summarized in Table 2. The psychological impact was compared between Group 1 and 2. Among the 74 patients whose surgery was postponed, 34 patients postponed it themselves, while 40 patients’ surgeries were postponed by the cataract center. Among the 34 patients, the majority (88%) of them were concerned about COVID-19, while a minority (12%) could not attend due to mandatory quarantine measures due to travelling abroad and other reasons. For the 40 patients whose surgery was cancelled by the cataract center, this was a decision made by hospital management at the height of the pandemic.

Regarding the result of our survey, according to GAD-7, five patients had mild symptoms and two patients had severe symptoms of anxiety, whereas regarding PHQ-9, none of the patients reached the cutoff score for major depression.

When considering the GAD-7 scores, there was no statistically significant difference in comparing the symptoms of anxiety in Group 1 and Group 2 (*p*-value = 0.757). When comparing patients who postponed the surgeries themselves those who had them postponed by the cataract center, there was again no statistically significant difference in their GAD-7 scores (*p*-value = 0.816). Depression symptoms according to PHQ-9 could not be compared as no patients reached the cutoff score. Subgroup analysis (Table 3) was also performed but did not show differences in anxiety symptoms according to GAD-7, in terms of gender (male versus female) (*p*-value = 0.215), age (≥70 versus <70 years old) (*p*-value = 0.366), pre-operative Snellen visual acuity (≥0.1 versus <0.1) (*p*-value = 0.405), or frequency of washing hands (≥10 versus <10 times/day) (*p*-value = 0.484).

The results regarding knowledge and practices towards COVID-19 are summarized in Table 4. In essence, most patients (99%) were aware of the main routes of transmission of COVID-19 and practiced good personal hygiene. Moreover, 65.7% of patients washed their hands 10 times or more each day. Among the 34 patients who postponed their surgeries themselves, 22 (65%) washed their hands 10 times or more each day and 14 (41%) never rubbed their eyes. However, there was no statistical difference regarding the psychological impact with respect to their knowledge of COVID-19 and personal hygiene.

## 4. Discussion

The cancellation of elective cataract surgeries did not cause a significant psychological impact on our patients as compared to those with surgeries performed as scheduled. The majority of the patients (>80%) did not experience symptoms of depression and anxiety. None of the patients reached the cutoff for major depression, and of those who reported anxiety symptoms, the severity was minimal. The questionnaire results indicated that our sample population was relatively not distressed. In addition, since the postponement was due to a well-understood reason, namely the COVID-19 pandemic, and this was likely well within their expectations, a change in anxiety or depressive symptoms was not apparent in these patients. These findings are consistent with another UK study that showed that the pandemic did not affect patients’ decisions to attend the hospital for cataract surgery, as 83.3% indicated their willingness to attend their surgery [19].

On the other hand, Beale et al. reported that a moderate frequency of handwashing of six to ten times a day was associated with a lower risk of human coronavirus infection in a pre-COVID-19 cohort and advocated for the important role of handwashing in COVID-19 [20]. In comparison, 65% of the patients in the present study had even higher levels of hand hygiene (>10 times per day). Given that COVID-19 appeared to have a similar transmission mechanism to seasonal coronavirus [21,22], patients in the present study demonstrated adequate personal hygiene. However, we did not observe an obvious link between higher levels of anxiety and high-frequency handwashers in our cohort.

We acknowledge that the present study has its potential limitations. As the questionnaires were conducted after the cataract operations, this could contribute to the alleviation of anxiety or depressive symptoms. Furthermore, the patients recruited in this study were mostly elderly. Older adults might experience fewer life events and thus be less distressed than younger age groups [23,24]. Studies have shown that the perceived likelihood of contracting COVID-19 and anxiety is significant at younger ages but not significant at older ages [25]. Furthermore, the patients in the present study were of Chinese origin, and the results cannot be extrapolated to countries with different ethnicities and cultures.

## 5. Conclusions

In conclusion, a prospective observational study was performed to review the psychological impact of the cancellation of surgeries. Although our study showed that there was no significant psychosocial impact of the cancellation of cataract surgeries due to COVID-19, the ongoing emergence of COVID-19 variants will lead to further cancellations and rescheduling of elective surgeries. This work adds to the growing literature on COVID-19 that can help to direct healthcare policy and serves as a reminder that the delaying of cataract surgeries may have an impact on patients’ physical and mental health.

## Figures and Tables

**Table 1 ijerph-19-14852-t001:** Patient demographics.

Patient Demographics	
Questionnaire response	99
Number of men	56
Number of women	43
Age range	55–89 years old
Cataract operation cancelled/postponed due to COVID-19 outbreak	74 (74.7%)
Cause of postponement Self-postponed Postponed by cataract center	34 (45.9%) 40 (54.1%)

**Table 2 ijerph-19-14852-t002:** Results of GAD-7 and PHQ-9.

GAD-7 Score					
Minimal (0–4)	Mild (5–9)		Moderate (10–14)		Severe (15–21)
92	5		0		2
PHQ-9					
No major depression (<10)			Major depression (≥10)		
99			0		
GAD-7 score comparison between Group 1 and Group 2					
GAD-7 score		Group 1 (surgery postponed) (total: 74)		Group 2 (surgery not postponed) (total: 25)	
Minimal (0–4)		68		24	
Mild (5–9)		5		0	
Moderate (10–14)		0		0	
Severe (15–21)		1		1	
*p*-value: 0.757					
PHQ-9 score comparison between Group 1 and Group 2					
PHQ-9 score		Group 1 (surgery postponed) (total: 74)		Group 2 (surgery not postponed) (total: 25)	
No major depression (<10)		74		25	
Major depression (≥10)		0		0	

**Table 3 ijerph-19-14852-t003:** Subgroup analysis.

Subgroup Analysis		
GAD-7 score	Surgery postponed by patient themselves (total: 34)	Surgery postponed by cataract center (total: 40)
Minimal (0–4)	31	37
Mild to severe (5–21)	3	3
*p*-value: 0.816		
GAD-7 score	Male (total: 43)	Female (total: 56)
Minimal (0–4)	42	50
Mild to severe (5–21)	1	6
*p*-value: 0.215		
GAD-7 score	Age <70 (total: 26)	Age ≥70 (total: 73)
Minimal (0–4)	23	69
Mild to severe (5–21)	3	4
*p*-value: 0.366		
GAD-7 score	Pre-operative Snellen visual acuity < 0.1 (total: 13)	Pre-operative Snellen visual acuity ≥ 0.1 (total: 86)
Minimal (0–4)	12	80
Mild to severe (5–21)	1	6
*p*-value: 0.405		
GAD-7 score	Frequency of washing hands <10 times/day (total: 34)	Frequency of washing hands ≥ 10/day (total:65)
Minimal (0–4)	32	60
Mild to severe (5–21)	2	5
*p*-value: 0.484		

**Table 4 ijerph-19-14852-t004:** Patients’ knowledge and practices toward COVID-19.

Which of the following is/are the route(s) of transmission of COVID-19?	
Eyes	2 (2%)
Respiratory tract	63 (63.6%)
Gastrointestinal tract	12 (12.1%)
All of the above	35 (35.3%)
How often do you wash your hands each day?	
0–3 times	3 (3%)
4–6 times	15 (15.21%)
7–9 times	16 (16.2%)
10 times or more	65 (65.7%)
How often do you rub your eyes?	
Never	47 (47.5%)
Occasional	49 (49.5%)
Always	3 (3%)

## Data Availability

Not applicable.

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
