# Peer review of "Psychological Impact of Cancellation of Elective Surgeries for Ophthalmic Patients during COVID-19 Pandemic"

_ijerph, 2022, doi:10.3390/ijerph192214852_

Round 1
Reviewer 1 Report
The article “Psychological impact of cancellation of elective surgeries for 2 ophthalmic patients during COVID-19 pandemic” need to be improved.
Authors should clarify these aspects.
The introduction, discussion and conclusions should be significantly improved.
Published articles on this topic need to be referenced in the introduction. There are published articles and at least one report from an academy of ophthalmology that relate or mention COVID, cataract patients and ophthalmology professionals. Also look for articles that relate COVID to other ophthalmologic pathologies.
In the discussion, a comparison should be made with other more or less related studies within ophthalmology.
I believe that this article is not of great scientific interest if it is not enriched with data such as those requested.
Author Response
Please see the attachment. Thank you.
Reviewers, IJERPH
RE: ijerph-1930721, entitled "Psychological impact of cancellation of elective surgeries for ophthalmic patients during COVID-19 pandemic”.
Thank you for your reply and kind consideration of our manuscript for publication. Enclosed please find a revised version of the manuscript, and the text below addressing the suggestions of the reviewer for your kind consideration.
Our reply to the reviewer’s comments is listed out in the following
- The introduction, discussion and conclusions should be significantly improved
- The introduction, discussion and conclusions have been re-written.
- More background with reference to previous ophthalmological journals on COVID-19 and reasons behind the objectives of the study are made and explained in the introduction.
- Published articles on this topic need to be referenced in the introduction. There are published articles and at least one report from an academy of ophthalmology that relate or mention COVID, cataract patients and ophthalmology professionals. Also look for articles that relate COVID to other ophthalmologic pathologies.
- We have included more references from ophthalmological literatures that relate COVID, such as European literatures and another study conducted in Saudi Arabia that relate COVID to other ophthalmologic pathologies such as glaucoma.
- In the discussion, a comparison should be made with other more or less related studies within ophthalmology
- Although there are no other studies that study the psychological impact on cancellation of elective ophthalmological surgeries, we have included a study conducted in UK, which showed the pandemic did not affect the patients’ willingness to come for cataract surgeries.
- We have also included more references that link COVID-19 and anxiety in terms of age.

Reviewer 2 Report
The authors aimed to identify the psychological impact of canceling cataract operations on a sample of Chinese patients.
This was an interesting paper, but I had some concerns about the methodology and analyses that limit my enthusiasm.
Introduction
1. The Introduction does not provide sufficient background and the reasons behind the objectives of the study are poorly explained.
Methodology
1. There appears to be a confused methodology. Indeed, the questionnaires that the authors used to detect anxiety and health in patients are slightly described in the Results section although it would be more adequate if they were described in Methods section.
2. Data analysis is not accurately described and it is not clear which statistical analysis the authors performed to verify their hypotheses. In addition, the authors should have devoted a paragraph exclusively to data analysis.
Results
1. The results are confusing and they are not adequately supported by tables.
Author Response
Please see the attachment. Thank you.
Reviewers, IJERPH
RE: ijerph-1930721, entitled "Psychological impact of cancellation of elective surgeries for ophthalmic patients during COVID-19 pandemic”.
Thank you for your reply and kind consideration of our manuscript for publication. Enclosed please find a revised version of the manuscript, and the text below addressing the suggestions of the reviewer for your kind consideration.
Our reply to the reviewer’s comments is listed out in the following
- The Introduction does not provide sufficient background and the reasons behind the objectives of the study are poorly explained.
- We apologize that the background and objectives of the study are poorly explained. We have rewritten the introduction - more references to ophthalmological journals on COVID-19 and explained our the purpose of our study.
- There appears to be a confused methodology. Indeed, the questionnaires that the authors used to detect anxiety and health in patients are slightly described in the Results section although it would be more adequate if they were described in Methods section.
- After revision, the questionnaires are described in Methods section.
- Data analysis is not accurately described and it is not clear which statistical analysis the authors performed to verify their hypotheses. In addition, the authors should have devoted a paragraph exclusively to data analysis.
- Thank you for pointing out. We have included a data analysis paragraph and the statistical analysis is described in methods section.
- The results are confusing and they are not adequately supported by tables
- We have revised the tables in the revised manuscript to summarize the results.

Round 2
Reviewer 1 Report
Accept in present form
Reviewer 2 Report
I am satisfied with the revisions made by the authors.